# *PHOENIX*: Photonic Distillation Transfers Electronic Knowledge to Hybrid Optical Neural Networks

## Abstract

As artificial intelligence (AI) systems continue to scale in both complexity and dataset size, conventional electronic hardware faces significant challenges in meeting the demands of low-latency, high-throughput, and energy-efficient processing, particularly for industrial deployments. However, sustaining such scaling is increasingly constrained by the physical and energy limitations of electronic computing. Optical Neural Networks (ONNs), leveraging the superior physical properties of photons, offer inherent advantages such as ultra-fast processing speed, massive parallelism, and near-zero power consumption, which have already demonstrated potential on simple tasks in small datasets like MNIST classification. In this work, we presented the first optoelectrically fused neural network deployment framework (*PHOENIX*) for object detection tasks, demonstrating its performance in industrial-level large datasets (e.g., COCO) and benchmark models. Compared to state-of-the-art electronic models, our solution achieved approximately **85.0**% accuracy. The accuracy was further improved to **93.0**% through our novel knowledge distillation strategy. Furthermore, we achieved **72.6**% energy reduction and **11.3×** speed acceleration compared to equivalent edge GPUs by successfully transferring spatial attention knowledge from the electronic domain to the photonic domain, making it an ideal choice for real-time, energy-critical industrial applications. This technique not only bridges the performance gap but also offers an alternative physically interpretable platform for AI. Our universal framework paves the way for extending ONN deployment to a wider range of deep learning models and applications, whether based on CNN or Transformer architectures, providing a compelling choice for real-time, energy-critical scenarios such as autonomous driving, smart surveillance, and industrial automation. Source code is available at https://github.com/Anon-BOTs/Distill-Hybrid-Optoelectronic

## 1 Introduction

Over the past decade, Artificial Intelligence (AI) has penetrated diverse fields, with computer vision (CV) standing out as one of its most transformative applications, revolutionizing industries ranging from healthcare to the robotics field, e.g., autonomous driving. Deep learning technology with Convolution neural networks (CNNs) (LeCun et al., 1998) and transformer-based (Vaswani et al., 2017) architectures (Figure 1.(ii)), represent a significant successful application in image processing. However, these deep learning model advancements heavily rely on digital electronic chips, which demand substantial computational resources and energy consumptions (Strubell et al., 2020). As models grow in complexity, the von Neumann architecture and conventional silicon-based hardware face inherent bottlenecks in memory bandwidth and power efficiency, limiting their scalability for real-time edge-side deployment.

In recent years, optical neural networks (ONNs) have attracted significant research attention for inference tasks such as the object classification task (Meng et al., 2023; Gu et al., 2021; Lin et al., 2018) (Figure 1.(a.i)), which offer a compelling solution by harnessing the inherent advantages of photonics: ultra-low latency, high parallelism, and energy-efficient linear operations enabled by the physical properties of light (e.g., coherence, light speed, and transmission as computing). Recent advances in integrated photonic circuits and diffractive optical networks (Lin et al., 2018)

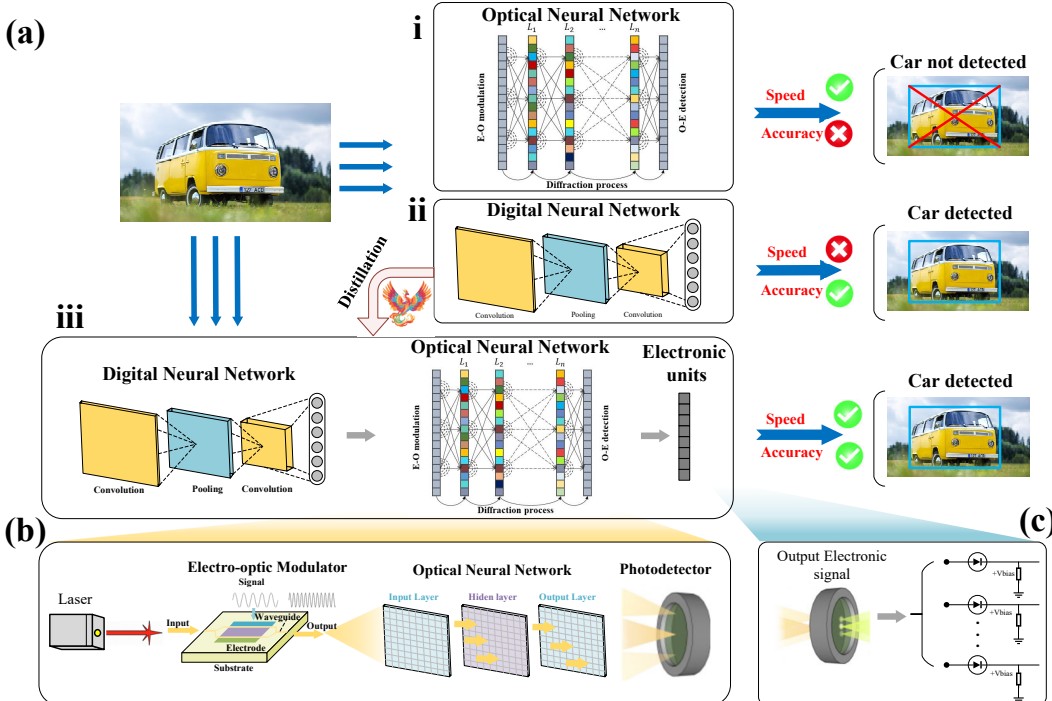

Figure 1: Integration of Optical Neural Networks into Computer Vision Tasks. (a) i) Optical neural network; ii) Digital neural network; iii) Distilled electronic–optical neural network. (b) Detailed components of the optical neural network pipeline. (c) Detailed components of the electronic sub-network.

have demonstrated ONNs' potential for matrix multiplication and inference tasks at speeds orders of magnitude faster than electronic counterparts (Clements et al., 2016; Feldmann et al., 2021). Nevertheless, despite their theoretical promise, prior works on ONNs have been largely confined to lower computer vision tasks, which focused on smaller datasets and simple classification tasks (e.g., MNIST (LeCun et al., 1998)) (Lin et al., 2018; Fu et al., 2023; Shen et al., 2017).

However, optical computing systems face several intrinsic challenges. First, analog signal processing introduces noise accumulation and lacks robust non-linear activations, limiting expressiveness. Second, training often relies on expensive full-wave optical simulations and gradient computations (Nikkhah et al., 2024a). Third, current models have poor generalizability, as training and deployment are typically tailored to specific hardware or datasets, with limited use in real-world CV tasks such as object detection (De Marinis et al., 2019). These issues underscore the need for more efficient training methods and co-designed optical-electronic solutions.

In this paper, we presented the *PHOENIX*: a hybrid optoelectronic neural network deployment framework. Inspired by distillation methods, this tool effectively transfers knowledge from well-trained electronic teacher networks (Figure 1.(a.ii)) to optical student architectures. By combining the ultra-low latency and energy efficiency of ONNs with the robust feature extraction capabilities of electronic CNNs, *PHOENIX* overcomes the inherent limitations of ONNs in learning nonlinear representation. Based on the *PHOENIX*, we successfully *applied ONNs to large-scale image object detection tasks for the first time* (Figure 1.(b)). Our approach not only bridges the gap between optical computing and practical CV applications but also establishes a framework for optimizing ONNs via distillation. The results underscore the potential of ONNs as a hardware-accelerated solution for next-generation AI, combining the speed of light with the rigor of machine learning.

The main contributions of our work are summarized as follows:

- *PHOENIX—The first deployment tool specifically tailored for the optoelectronic heterogeneous neural networks:* *PHOENIX* transferring knowledge from electronic "teacher" models to efficient ONN "student" architectures, enhancing robustness against optical noise and hardware variations, achieving **93%** accuracy compared to the Electronic model.

- **A pioneering instance of photonic deployment for an industrial-scale task:** We introduced ONNs into industrial-scale object detection tasks (e.g., COCO) , advancing beyond prior small-scale ONN applications (e.g., MNIST), marking a significant step toward practical ONN deployment.

- **A universal, energy-efficient, low-latency optoelectronic heterogeneous hardware deployment architecture:** Our general framework is compatible with both CNN-based or Transformer-based architectures. Employing a hardware-aware co-design approach, we achieved significant improvements in latency and power consumption compared to electronic counterparts. The latency ranges from 8 ms for CNN to 20.8 ns for ONN, resulting in a 50% reduction in total inference time and an 86.5% energy reduction. This enables ultra-low-power, high-speed operation for edge devices and promotes sustainable AI development.

## 2 RELATED WORK

### 2.1 OPTICAL NEURAL NETWORKS (ONNs)

Optical Neural Networks (ONNs) have emerged as promising alternatives to traditional electronic-based deep learning architectures. Using optical phenomena for computation, ONNs naturally leverage advantages such as ultralow latency, massive parallelism and lower energy consumption (Hua et al., 2025; Wetzstein et al., 2020; Xu et al., 2024a) compared to their electronic counterparts. Current ONN implementations are primarily divided into two categories: (i) Diffractive structures employ sequential free-space diffraction and optical modulation through spatial light modulators (SLM) with computational weights encoded in the phase profile. (ii) On-chip architectures utilize optical components such as Mach-Zehnder interferometers (Shen et al., 2017), microring resonators or phase-change materials (Wang et al., 2024; Wu et al., 2021) to enable compact optical information processing. To date, ONNs have successfully demonstrated their potential in matrix multiplications, convolution operations and solving simple integral equations (Nikkhah et al., 2024b; Xu et al., 2021; Cordaro et al., 2023). However, the intrinsic lack of optical nonlinearities and limited tunable parameters restricts ONNs to small-scale computations or low-level AI tasks such as vowel recognition or MNIST classification (Shen et al., 2017; Lin et al., 2018). A number of recent works have shifted attention toward optoelectronic hybrid systems, recognizing their promise in bridging optical and electronic computing (Kissner et al., 2024; Shiflett et al., 2021; 2023). Optoelectronic hybrid systems combine electronic flexibility with optical computing potential but face dual bottlenecks: (1) Hardware requires additional components and incurs energy costs from optoelectronic conversions. (2) Algorithmically, scalable training methods remain underdeveloped for matching electronic neural network performance in complex tasks.

### 2.2 DEEP LEARNING FOR OBJECT DETECTION & KNOWLEDGE DISTILLATION FOR EFFICIENT MODELS.

Object detection is a fundamental problem in computer vision and has witnessed significant advancements over the past decade, primarily driven by deep learning techniques. Early works of object detection have evolved from two-stage CNN architectures like Faster R-CNN (Girshick, 2015) to single-stage detectors such as YOLO (Redmon et al., 2016) and anchor-free approaches like FCOS (Tian et al., 2019). Transformers further advanced the field, with Swin Transformer (Vaswani et al., 2017) and DINO (Zhang et al., 2022) achieving state-of-the-art results. However, these models rely on computationally expensive operations (e.g., attention mechanisms), making them impractical for edge deployment. Knowledge distillation is a widely adopted model compression and optimization technique introduced (Buciluă et al., 2006) and further popularized by (Hinton et al., 2015), later extended to object detection (Chen et al., 2017). It transfers knowledge from a larger "teacher" network to a smaller "student" network, significantly reducing the computational cost while maintaining or even enhancing accuracy. While most distillation approaches have traditionally focused on electronic neural networks. The concept has rarely been explored in the context of ONNs (Xiang et al., 2022; Wirth-Singh et al., 2024), all of this work focuses on simple tasks like image classification on a small-scale dataset (e.g., MNIST, FashMNIST). This study pioneers challenging the complex image object detection tasks on a large-scale dataset, which applies knowledge distillation into Optical Neural Networks, demonstrating substantial benefits in performance and efficiency.

Figure 2: Overview of the proposed hybrid photoelectronic object detection framework, **PHOENIX**. The system is built upon a state-of-the-art baseline detector, where early-stage extracts low/mid-level features by the CNN-based or Transformer-based method. The key component of the knowledge distillation module transfers all-electronic 'teacher' backbone to the ONN 'student' stages, enhancing their functional capabilities. Finally, features output by the ONN-processed segment of the backbone are fed to a task-specific detection head for object classification and localization.

## 3 METHODOLOGY

In this section, we present our framework **PHOENIX**, the first **general knowledge distillation architecture** that integrates Optical Neural Network (ONN) modules, which leverage the speed and energy efficiency of photonic computing in the complex task of object detection within computer vision. We detail the architecture in Figure 2 and present the process of the Optical Neural Networks (ONN) in Section 3.1, the optoelectronic integration strategy in Section 3.2, the knowledge distillation technique in Section 3.3 and the final detection process in Section 3.4.

### 3.1 OPTICAL NEURAL NETWORKS (ONNS)

Optical Neural Networks (ONNs) leverage optical components such as interferometers, modulators, and photonic circuits to perform computations traditionally carried out by electronic processors. The main advantage of ONNs over conventional electronic neural networks is their inherent parallelism and speed, which allows them to handle matrix-vector multiplications at ultra-low latency while consuming much less power compared to electronic models.

**ONN architecture** The architecture of our ONN model and its associated mathematical formulation of light propagation are illustrated in Figure 3. The forward propagation process in a diffractive optical neural network can be systematically divided into two main stages: propagation and modulation. During propagation, each pixel acts as a secondary source, emitting a wave that transmits its signal to the subsequent layer. As the light advances, it encounters the modulation stage, where its characteristics are altered, either in phase or amplitude, depending on the nature of the modulation layer. The underlying physical model, which is particularly vital for on-chip implementations, is governed by the rigorous Rayleigh-Sommerfeld diffraction equation (Goodman, 2005). Such approach dispenses with the need for restrictive far-field

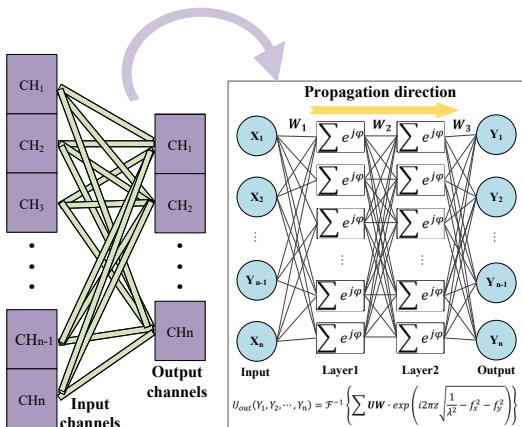

Figure 3: The architecture pipeline and mathematical modeling of the ONN module.

or small-angle approximations, thereby ensuring a more faithful representation of light propagation. In this framework, every pixel in a given layer, located at coordinates $(x_i, y_i, z_i)$, serves as the origin of a secondary wave. The propagation of this wave between adjacent layers is precisely described by

the following equation:

$$w_i^l(x, y, z) = \frac{L}{r^2}\left(\frac{1}{2\pi r} + \frac{1}{j\lambda}\right)\exp\left(j\frac{2\pi r}{\lambda}\right).$$ (1)

Here, $w_i^l(x, y, z)$ denotes the contribution of a single neuron, which can be regarded as a secondary wave source. The parameter of subscript $i$ refers to the pixel index within the $l^{th}$ propagation layer, $\lambda$ represents the wavelength of the light, and $L$ specifies the axial distance between adjacent layers, while $k$ designates the index of the subsequent layer. The term $r = \sqrt{(x - x_i)^2 + (y - y_i)^2 + (z - z_i)^2}$ defines the Euclidean distance between two pixels, with $(x, y, z)$ and $(x_i, y_i, z_i)$ denoting their respective spatial coordinates. Furthermore, $j = \sqrt{-1}$ is the standard imaginary unit.

In this work, we employ a simplified diffractive model as a representative example, noting that it can be *substituted with any ONN architecture* possessing sufficient representational capacity. The choice of the diffractive model stems from its inherent similarities to convolution operations. Free-space diffraction fundamentally constitutes a convolution with an impulse response function, while phase/amplitude modulation can be conceptually equivalent to a convolution kernel with specific receptive field characteristics. To implement convolution layers with arbitrary input-output channel configurations, we utilize $N_{in} \times N_{out}$ optical network units to construct a channel-combining convolution layer. This configuration can be simplified through strategic reuse of fundamental optical network modules and the segmentation of effective optical field.

### 3.2 HYBRID OPTOELECTRONIC OBJECT DETECTION NEURAL NETWORKS

The key innovation in our general photoelectric object detection framework is that: (1) maintains electronic processing in early stages for robust feature extraction, (2) selectively offloads computationally intensive operations to photonic hardware in deeper stages. Notably, our framework is *compatible with any detection architecture*. In this work, we demonstrate its effectiveness using both RegNet (Radosavovic et al., 2020) and ViT-Base (Dosovitskiy et al., 2020) as backbones, with RegNet serving as the primary example in the following sections.

#### 3.2.1 BASELINE OBJECT DETECTION ARCHITECTURE

Our framework builds upon FCOS (Tian et al., 2019), which is a fully convolution one-stage object detector that predicts objects by regressing bounding boxes and classification scores directly at each feature map location. Taking the **RegNet-Y** as the backbone, an initial stem layer followed by **four main stages** (Stage 1, Stage 2, Stage 3, Stage 4). Each stage comprises *a sequence of bottleneck blocks* that progressively reduce spatial resolution while increasing channels and learning hierarchical features, the architecture is detailed in Appendix.A.2. Actually, our framework is suitable for any detection approaches, not limited to these four-stage methods.

#### 3.2.2 HYBRID CNN-ONN ARCHITECTURE

**Electronic**   To integrate optical computing into the object detection pipeline, we design a hybrid backbone architecture commences with an initial electronic segment, $M_A$, where typically comprises the first stage (or stages 1 to $k$, where $k \in \{1, 2, 3\}$) of the RegNet backbone, responsible for processing the raw input image data. These CNN layers capture low and mid-level features of image, such as precise edge and texture information, maintain compatibility with standard vision pre-processing. However, the raw high-dimensional image data for these initial tasks will be suboptimal if we directly apply ONNs, due to challenges in implementing identical non-linear optically with high fidelity or without significant energy for active optical elements. Therefore, by first processing the input with this robust electronic segment, we extract and refine a rich set of features that are more amenable and efficiently processed by the subsequent ONN modules (segment $M_B$), effectively preparing the data for the strengths of optical computation.

**Hybrid**   To harness the advantages of photonic computation, we selectively replace the computationally intensive later stages (stages 2 to 4) of the RegNet-Y backbone with ONN modules. After the electronic layer process, the deeper feature abstraction and semantic representation are offloaded to the ONN for acceleration and energy-efficient processing, as the ***Hybrid Model*** part shown in

Figure 2. The ONN module is implemented using a free-space diffractive optical network configured to approximate the convolution layers via Fourier optics, the detailed architecture as illustrated in Section 3.1. In summary, our optoelectronic integration strategy offers flexibility in partitioning the backbone. The hybrid backbone is a sequential composition of an initial electronic segment, $M_A$, processed by CNNs, and a subsequent photonic segment, $M_B$, processed by ONNs. This process can be represented as follows:

$$M_{\text{hybrid}} = M_{A,\text{Stages } 1...k} \bigotimes M_{B,\text{Stages } k+1...N} \quad (2)$$

Here, '$\bigotimes$' denotes sequential processing. The electronic segment $M_{A,\text{Stages } 1...k}$ comprises the initial $k$ stages of the RegNet-Y backbone to extract lower-mid level features, which can be chosen from $\{1, 2, 3\}$, allowing for different configurations of initial electronic processing. Specifically, k=4 denotes that the framework with a fully CNN model without ONNs. Consequently, the photonic segment $M_{B,\text{Stages } k+1...N}$ consists of the remaining stages, from Stage $k + 1$ up to Stage $N$ (where $N$ is the total number of main stages in RegNetY, typically 4).

## 3.3 KNOWLEDGE DISTILLATION FOR ONN NON-LINEARITY ENHANCEMENT

A significant challenge for ONNs is effectively implementing complex non-linear activation functions and learning the intricate non-linear mappings achieved by deep electronic networks. To address this, we employed a knowledge distillation strategy, the architecture present in Figure 4. This process occurs during the training of the ONN components and involves the following two main models:

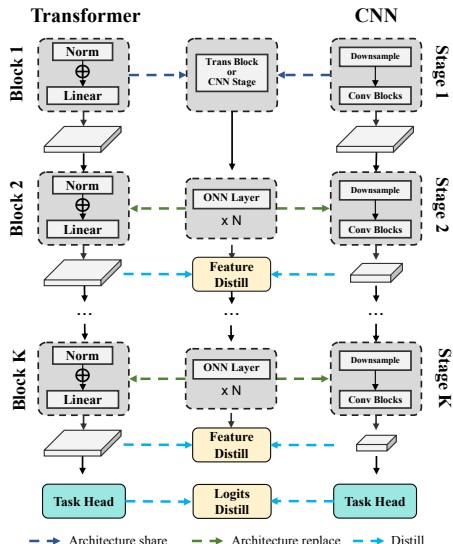

Figure 4: Distillation architecture for ONNs. The teacher model is flexible for any CNN-based and Transformer-based approaches.

- **Teacher Model:** A fully electronic, pre-trained baseline FCOS with CNN-based or Transformer-based backbone in object detection dataset, which provides the learned feature representations for better understanding of images to the student.

- **Student Model:** A series of ONN modules shown in Section 3.1, segment of the backbone $M_B$ is realized as the student model. These ONNs leverage distinct optical attributes (e.g., high-speed linear processing, unique analog responses) and capture the nonlinear ability from the teachers.

The goal of distillation is to transfer the "knowledge" from the teacher's electronic backbone (corresponding to those in $M_A$) to the student's $M_B$ segment. This is achieved by encouraging the output feature maps of the ONN modules in the student network to mimic the output feature maps of the corresponding electronic stages in the teacher network. The distillation loss, $\mathcal{L}_{KD}$, can be formulated as the Mean Squared Error (MSE) loss between the feature maps:

$$\mathcal{L}_{KD} = \sum_{i \in S_{M_B}} \lambda_i \cdot \frac{1}{N_i} \sum_{j=1}^{N_i} (f_{T,j}^i(x) - f_{S,j}^i(x))^2 + L_{kl}(P_T, P_S) \quad (3)$$

where $P_T$ and $P_S$ are the output logits of the teacher and student model, respectively. $L_{kl}$ denote the Kullback-Leibler Divergence and $S_{M_B}$ is the set of backbone stages within the $M_B$ segment (i.e., stages $k + 1$ to $N$), $\lambda_i$ are weighting factors for the $i$-th stage, $f_{T,j}^i(x)$ and $f_{S,j}^i(x)$ represent the $j$-th element of the flattened feature maps from the teacher and student networks at stage $i$ for input $x$, respectively, and $N_i$ is the total number of elements in the feature map for stage $i$ (e.g., $N_i = H_i \times W_i \times C_i$ for feature maps with spatial dimensions $H_i \times W_i$ and $C_i$ channels). The inner summation calculates the squared error element-wise, which is then averaged by dividing by $N_i$ to compute the MSE for that stage. By training the $M_B$ (ONN) portions of the student network to replicate the behavior of the highly teacher-staged, we effectively imbue the ONNs with enhanced non-linear processing capabilities.

### 3.4 DETECTION HEAD AND LOSS FUNCTION

The distilled photoelectric features from the distillation process in Section 3.3 through the ONN modules and then fed into a task-specific detection head which operates in a fully convolution manner, making dense, per-pixel predictions across the spatial dimensions of the feature maps. During training stage, we minimize the total loss, which consists of the distillation loss $L_{kd}$ and the detection loss $L_{det}$, performs simultaneous classification and bounding box regression : $\mathcal{L}_{det} = \mathcal{L}_{cls} + \lambda_{bbox}\mathcal{L}_{bbox}$.

$$\mathcal{L}_{total} = \lambda_{kd}\,\mathcal{L}_{KD} + \lambda_{det}\,\mathcal{L}_{det} = \lambda_{kd}\,\mathcal{L}_{KD} + \mathcal{L}_{cls} + \lambda_{bbox}\mathcal{L}_{bbox} \qquad (4)$$

At inference time, the trained model processes the input image, predicts bounding boxes and labels in real-time with low power consumption, leveraging the optical layers.

## 4 EXPERIMENTS

### 4.1 EXPERIMENTS SETUP

**Dataset & Models**   We evaluated our proposed hybrid Optoelectronic object detection framework on the MS COCO 2017 dataset (Lin et al., 2014), a standard benchmark for large-scale object detection. The dataset contains over 118,000 training images and 5,000 validation images, covering 80 object categories. Performance was evaluated using the standard mean Average Precision (mAP) metric at IoU thresholds from 0.5 to 0.95 (mAP@[.5:.95]). We further evaluate our approach on (Caesar et al., 2020), a large-scale outdoor autonomous driving dataset to demonstrate the effectiveness and generalizability (see Appendix.A.3). Our implementation was verified on various backbones, e.g., RegNet and ViT-Base. All models were trained on NVIDIA A800 GPUs, details can be found in the Appendix.A.1. The optical simulation method employed in our work is grounded in established practices widely used in the EDA and computer architecture communities (Binkert et al., 2011; Zhou et al., 2021; Xu et al., 2024b), ensuring the credibility of latency, area, and energy estimates. Our diffractive ONN simulation approach also has been experimentally validated in multiple peer-reviewed studies (Lin et al., 2018; Fu et al., 2023; Luo et al., 2019). Notably, the design parameters used in our model are derived from experimentally demonstrated systems in (Lin et al., 2018; Fu et al., 2023), reinforcing the practical relevance and feasibility of our architecture.

### 4.2 MAIN RESULTS ANALYSIS

Table 1: Detection Performance on MS COCO with image resolution of $677 \times 400$.

| Method | Setting | $AP_{50}$ | $AP_{75}$ | AP@[.5:.95] | Relative SOTA (%) |
|---|---|---|---|---|---|
| RegNet-Y | Baseline | 46.5 | 31.2 | 29.8 | 100% |
| | Hybrid-1 $\times layer$ ONN | 39.6 | 26.0 | 24.9 | **83.6%** |
| | Hybrid-2 $\times layer$ ONN | 30.5 | 17.5 | 17.5 | **58.7%** |
| | Hybrid-3 $\times layer$ ONN | 19.6 | 8.0 | 9.5 | **31.9%** |
| | Distill-1 $\times layer$ ONN | **41.6** | **27.3** | **26.3** | **88.3%** |
| ViT-Base | Baseline | 64.5 | 48.9 | 45.0 | 100% |
| | Hybrid-1 $\times layer$ ONN | 58.8 | 41.1 | 38.2 | **85.0%** |
| | Hybrid-2 $\times layer$ ONN | 47.0 | 28.5 | 27.8 | **61.8%** |
| | Distill-1 $\times layer$ ONN | **60.1** | **42.4** | **41.8** | **93.0%** |

#### 4.2.1 SCALABILITY COMPARISON WITH ELECTRONIC LEARNING PROTOCOLS

**Hybrid Optoelectronic Performance**   While the core motivation for exploring hybrid optoelectronic architectures lies in their potential for substantial efficiency gains—such as reduced latency and lower energy consumption due to the distinct *electrical attributes* of optical processing. The RegNet-Y and ViT-base model, a purely electronic implementation of CNN-base and Transformer-base architecture, achieves an mAP of 29.8% and 48.9%, serving as our reference, as shown in Table 1. These hybrid models aim to leverage the *optical attributes* of ONNs, such as ultra-speed parallel processing for linear operations (e.g., convolutions, matrix multiplications). However, we observe a progressive decline in AP as more layers are offloaded to the optical domain. "Hybrid-1

x layer ONN" (e.g., one RegNet-Y stage replaced) drops to an AP of 24.9 (83.6% of baseline), while "Hybrid-3 x layer ONN" (e.g., three RegNet-Y stages replaced) shows the most significant degradation with an AP of 9.5 (31.9% of baseline). This performance degradation can be attributed to several inherent differences between mature electronic computation and current practical ONN implementations. While ONNs excel at rapid linear transformations, they may not natively support the same range or precision of non-linear activation functions commonly used in CNNs. Furthermore, optical computations can be more susceptible to analog noise (e.g., thermal fluctuations, detector shot noise, fabrication imperfections in photonic circuits) and may operate at a lower effective numerical precision compared to 32-bit floating-point operations typical in electronic GPUs.

To further verify robustness, we conducted quantitative experiments under varying noise levels (0%, 3%, 5%, 10%). Results show that the mAP of the hybrid ONN drops modestly from 41.8% to 40.1% under 10% phase noise, while the KD-enhanced version retains 96% of the original hybrid result and 89% of the baseline accuracy (originally 93%). This confirms the resilience and robustness of our method under real-world conditions.

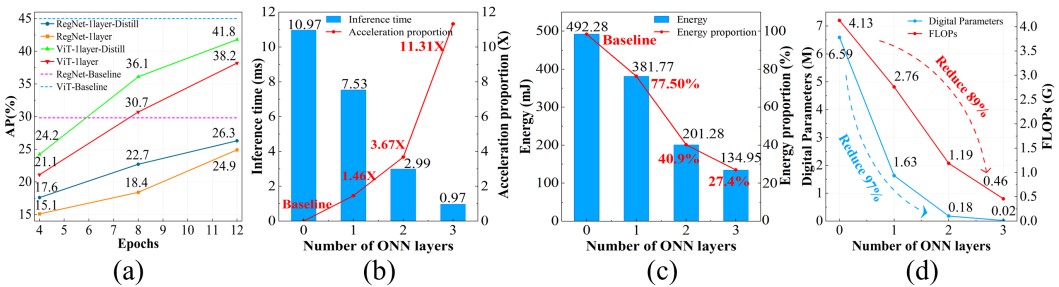

Figure 5: The performance across different numbers of ONN layers on : a) Different epochs. b) Acceleration of inference time. c) Energy consumption. d) FLOPs and parameters reduction.

**Knowledge Distillation**    The performance degradation in non-distilled hybrid models, as detailed above, directly substituting electronic layers with optical ones can therefore lead to a significant functional mismatch, stems from the inherent discrepancies between the electrical attributes of conventional CNNs or Transformers and the current optical attributes of ONN counterparts (potentially different non-linear responses, susceptibility to analog noise, and distinct precision characteristics). To address these, our distill approach transfers the "knowledge" from the teacher's electronic backbone to the student ONNs model. As shown in Table. 1, architecturally identical to worse performance "Hybrid-1 x layer ONN" but trained with our distillation strategy, it achieves an AP of 41.8, recovering performance to **93.0%** of the baseline. The performance of distilling various numbers of ONN layers at different epochs are also presented in Figure 5a. Summary all, the knowledge distillation serves as a powerful bridge, guiding the ONN student to achieve a functional outcome comparable to the sophisticated electronic teacher, which contributes the following features:**i) Compensates for Non-linearity Disparities, ii) Mitigates Impact of Lower Precision and Analog Noise, iii) Provides Effective Learning Signals in a Constrained Space**. More details analysis can be found in Appendix.A.2.4.

### 4.2.2 COMPUTATIONAL EFFICIENCY OF HYBRID OPTOELECTRONIC NETWORKS.

Optical computing offers several fundamental advantages over its electronic counterpart, particularly for operations prevalent in CNNs and Transformer architectures. Firstly, computations in the optical domain can, in principle, occur at the speed of light and optical systems exhibit massive intrinsic parallelism. For instance, a simple lens can perform a 2D Fourier transform on an entire image plane simultaneously, processing all pixels in parallel. This inherent parallelism is key to accelerating matrix-vector multiplications and convolutions. Secondly, power consumption can be drastically reduced. Passive optical components consume negligible power, and active components are continuously improving. A third critical advantage is scalability with input resolution. Certain optical processing schemes, such as Fourier optics-based convolution or systems utilizing a fixed Point Spread Function (PSF), offer a potential $O(1)$ computational complexity for the core optical operation with respect to the number of input pixels ($H \times W$). These fundamental advantages translate into significant

theoretical speedups for specific neural network operations when implemented on specialized optical neural network (ONN) hardware.

For instance, we show ***how the ONN module accelerates CNN computations to $10^5 \times$***. By replacing a 4-stage CNN with a 2-stage hybrid optoelectronic architecture incorporating two ONN layers, we reduced the electronic processing time by $T_{cnn} \approx 8ms$, shown in the Figure 5b. Specifically, our ONN photonic chip is composed of a laser source, an electro-optic modulator (EOM), a diffractive optical neural network (DONN), a photodetector (PD) and an array of Schottky barrier diodes (SBD). The temporal response of each component is denoted as $t_{laser}$, $t_{eom}$, $t_{onn}$, $t_{pd}$ and $t_{sbd}$, respectively. The EOM switching speed (4.46 ps) and PD response time (10.40 ns) are based on (Zhang et al., 2025; Wu et al., 2025). The contribution to the overall operational time of Schottky diodes and the laser is negligible due to the normally-open configuration, where $t_{sbd} = t_{laser} = 0$ s. $L$ represents the distance between adjacent layers, and $c$ represents the speed of light. Therefore, $t_{onn} = 3L/c = 3 \times 250\mu m/c \approx 2.50$ ps. The overall operational time for a single ONN stage is then determined by:

$$T_{onn} = t_{EOM} + t_{onn} + t_{PD} = 10.40 \ ns \tag{5}$$

By insteading of the electronic modules (CNNs) with ONNs. The efficient improves $T_{factor} = Tcnn/2 * T_{onn} = 8ms/20.8ns \approx 3.8 \times 10^5$ times, this represents a theoretical speedup of five orders of magnitude for the core optical operation itself. Such a component-level improvement underscores the transformative potential of ONNs and contributes significantly to the overall system-level speedups observed in hybrid architectures. Obviously, increasing Optical Neural Network (ONN) layers in our hybrid system dramatically reduces the framework inference time. The chart shows processing time dropping from approximately 11 ms for the all-electronic baseline (0 ONN layers, full CNNs) to just around 1.0 ms when 3 ONN layers are used, accelerating **11.3×** compared to the baseline. Our experimental investigation also assessed the impact of increasing Optical Neural Network (ONN) layer integration on energy consumption and model complexity, specifically concerning digital parameters and electronic FLOPs. Each ONN chip consists of one of laser, electro-optic modulator, photodetector, passive diffractive network, and a Schottky diode array. Based on research literature, we found that the total power consumption of each ONN layer is:

$$E_{total} = E_{EOM} + E_{PD} + E_{SDA} + E_{Laser} \tag{6}$$

Since a single laser can be shared across multiple diffractive layers in a multi-layer ONN system, only one laser is required regardless of the number of layers. As depicted in the Figure 5c and Figure 5d, incorporating up to 3 ONN layers reduces energy consumption to just 27.4% of the all-electronic baseline (dropping from approximately 492 mJ to 134.95 mJ). Concurrently, this strategy slashes the number of digital parameters by up to 97% (from 6.7M down to 0.1M) and reduces electronic FLOPs by up to 89% (from 4.1 GFLOPs to 0.4 GFLOPs).

In summary, by strategically offloading computationally intensive tasks from electronic to the optical domain, our hybrid optoelectronic networks can break through the efficiency bottlenecks faced by purely electronic systems. This makes them particularly promising for high-throughput, low-latency, and energy-critical applications in areas such as autonomous driving, real-time video analysis, and edge AI, aligning with the experimental latency improvements demonstrated in our work.

**Visualization**  Moreover, we visualize the distilled optical features by performing numerical simulations using Lumerical FDTD. As illustrated in Figure 6a, our optical neural network comprises an input layer, two intermediate diffractive layers (each with a distance of $L = 250, \mu$m), and an output layer. The phase modulation profiles of the diffractive layers are directly determined by the trained weights of the ONN model. Figure 6b shows the detailed phase distributions of the intermediate diffractive layers, along with the corresponding simulation results of light field propagation through all four layers of the ONN.

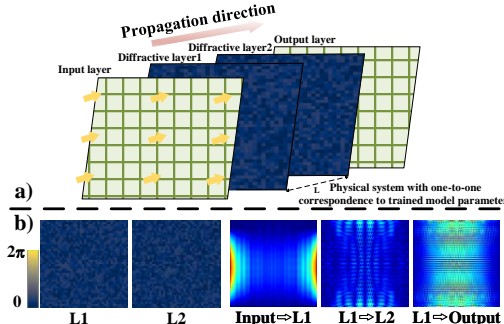

Figure 6: ONN experimental architecture and the Distribution of the Diffracted Field simulated in FDTD.

***LLM Assistance: We employed LLMs exclusively for text polishing; they were in no way involved in research ideation.***

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

# Appendix A

# Technical Appendices and Supplementary Material

In this section, we provide a supplementary material to support the findings presented in the main paper, including an extended dataset and training information, Section A.1, detailed network architectures, Section A.2, additional experimental results, Section A.3, and qualitative visualizations, Section A.4.

## A.1 DATASET AND TRAINING DETAILS

**nuScenes**  The nuScenes (Caesar et al., 2020) is a large-scale outdoor scene dataset and a comprehensive resource for autonomous driving research, which includes 700 scenes for training, 150 for validation, and 150 for testing. For our experiments, we utilized the full training set comprising approximately 28,130 samples and the validation set with 6,019 samples. Each sample includes data from 6 cameras, 1 LIDAR, 5 RADARs, GPS, and IMU. Our object detection task primarily focuses on the camera-based detection, e.g., cars, pedestrians, cyclists, using the provided 3D bounding box annotations projected into the 2D image planes. We follow the official train/validation split. Input images from nuScenes were employed 900x1600 pixels for processing. For the nuScenes 2D detection task, we take the same evaluation strategy as COCO's, with average precision and recall indicator.

**Implementation details.**  Our code implementation is based on the MMDetection3D (Contributors, 2020), all models in the main manuscript are trained for 12 epochs with batch size 1 on 32 A800 GPUs. We use Stochastic Gradient Descent (SGD) optimizer, with learning rate of $1 \times 10^{-2}$ for regnet and AdamW, with learning rate of $1 \times 10^{-4}$ for ViT-B, on COCO and nuScenes datasets, respectively. Specifically, we only use the ***front view of the camera***  in nuScenes dataset for training and evaluation. In this appendix, unless explicitly stated, the results are conducted on 8 A800 GPUs. For distillation, models with ONNs are trained in two stages: (1) pretraining the full electronic baseline as teacher model, (2) freezing teacher model and distilling feature of ONN-replaced blocks using feature and logits loss.

## A.2 NETWORK ARCHITECTURE AND PROCESS

This section outlines the specific configurations of the backbone architectures used in our experiments and presents the process of forward propagation in Optical Neural Network (ONN).

### A.2.1 VIT-BASE

For experiments on transformer-based backbones, we employed ViT-Base with Patch Size 16. The architecture consists of an initial patch embedding layer, stacked Transformer blocks, and a final layer norm which is shown in Figure 7.(a). Each Transformer block includes a multi-head self-attention (MHSA) module with 12 heads and a feed-forward network (FFN) with a 4 expansion and the model contains 12 transformer blocks, hidden size 768. Specifically, ONN modules replace the last N blocks during hybridization. We follow ViTDet (Li et al., 2022) to construct different level features for FPN.

### A.2.2 REGNET-Y

RegNet-Y (Radosavovic et al., 2020) is used as backbone for cnn-based model. It consists of a stem, followed by four stages, which has 1, 3, 7 and 5 blocks, respectively, which are shown as Figure 7.(b). Each block follows the standard bottleneck structure with group convolution. In addition, there will be a downsampling operation in each stage, and ONN module replaces CNN module after downsampling. Here, we present a 2-layer ONN hybrid framework configured as example in Table 2.

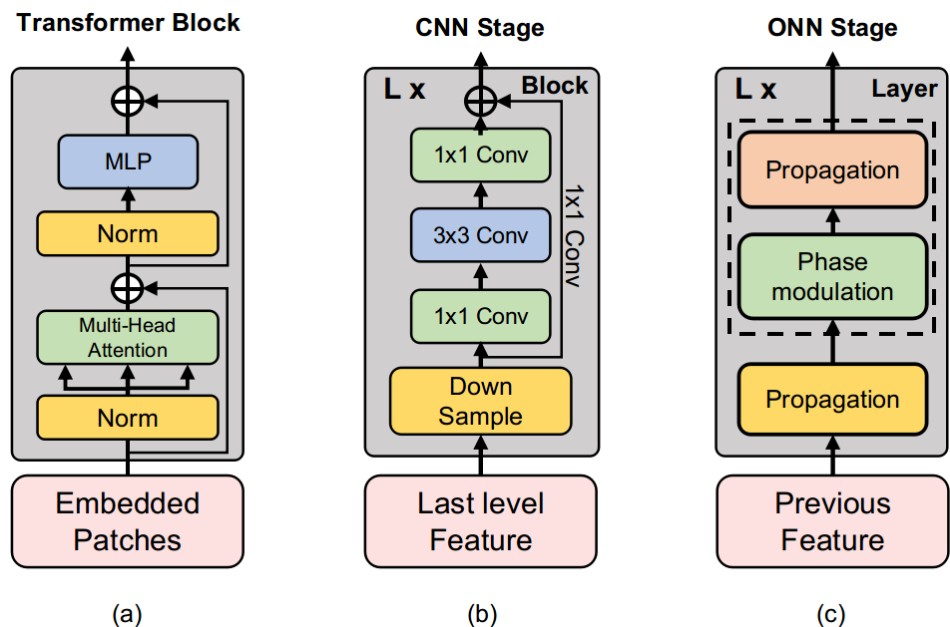

Figure 7: The architecture of ViT blocks (a), stages of RegNet (b) and ONN stages (c).

Table 2: RegNet-Y Backbone Architecture Specifications

| Stage | Channels | Depth | Stride | ONN Replacement |
|-------|----------|-------|--------|-----------------|
| S1 | 64 | 1 | 2 | ✓ (Electronic) |
| S2 | 128 | 3 | 2 | ✓ (Electronic) |
| S3 | 160 | 2 | 2 | ✓ (ONN) |
| S4 | 256 | 2 | 2 | ✓ (ONN) |

### A.2.3 OPTICAL NEURAL NETWORKS

We present the detailed architecture of ONN, which is shown in Figure 7.(c). It is mainly composed of the propagation module and the stacked ONN blocks in series. Furthermore, each block contains a propagation module and a phase modulation module.

**Forward propagation**  Optical Neural Networks (ONNs) are a new class of computational models that leverage optical components such as interferometers, modulators, and photonic circuits to perform computations traditionally carried out by electronic processors. The main advantage of ONNs over conventional electronic neural networks is their inherent parallelism and speed, which allows them to handle matrix-vector multiplications at ultra-low latency while consuming much less power compared to electronic models.

The incident field at the first layer, denoted as $\mathbf{U}_0$, subsequently propagates through a cascade of diffractive optical layers. This process culminates in the generation of the optical field at the output plane, designated as $\mathbf{U}_{N+1}$, which represents the final output of the network at the imaging plane (Zhou et al., 2020):

$$\mathbf{U}_{N+1} = \mathbf{W}_{N+1} \left( \prod_{k=N}^{1} \mathbf{M}_k \mathbf{W}_k \right) \mathbf{U}_0,$$

where, $\mathbf{U}_{N+1}$ represents the vectorized optical field at the $(N+1)^{\text{th}}$ layer, corresponding to the output plane. The matrix $\mathbf{W}_k$ denotes the diffractive weight matrix that models forward light propagation from the $(k-1)^{\text{th}}$ layer to the $k^{\text{th}}$ layer. In addition, $\mathbf{M}_k = \mathrm{diag}(e^{j\phi_k})$ describes a diagonal modulation matrix at the $k^{\text{th}}$ layer, where $\phi_k$ specifies the vector of phase modulation coefficients.

For a diffractive optical neural network that comprises $N$ intermediate layers, the detector positioned at the imaging plane measures the intensity distribution of the resulting optical field. This measured intensity then serves as the network's inference output, denoted by the following presentation $\mathbf{O}$:

$$\mathbf{O} = |\mathbf{U}_{N+1}|^2 = \left| \mathbf{W}_{N+1} \left( \prod_{k=N}^{1} \mathbf{M}_k \mathbf{W}_k \right) \mathbf{U}_0 \right|^2$$

### A.2.4 SUPPLEMENTARY FOR KNOWLEDGE DISTILLATION

The role of knowledge distillation is to act as a powerful bridge, guiding the ONN student to achieve functionality comparable to that of the sophisticated electronic teacher. The following features illustrate why this process is effective, which correspond to Section 4.2.1:

- **Compensates for Non-linearity Disparities**: The teacher network has learned complex data manifolds using its robust electronic non-linearity. While the student ONN, even if its native optical non-linearity are different or less expressive, is trained via KD to reproduce the output feature distributions or logit patterns of the teacher. This forces the ONN to configure its linear optical transformations and available non-linear mechanisms in such a way that the overall block-wise or stage-wise function approximates that of the teacher.

- **Mitigates Impact of Lower Precision and Analog Noise**: The teacher model often pre-trained on large datasets and possessing a well-designed architecture, has learned powerful inductive biases for the task and provides a "clean" high-precision target representation. By training the ONN student to match these target features, it encourages the student to learn representations that are robust enough to yield the correct output despite underlying analog noise or limitations in optical precision. The ONN learns to focus on the salient, high-signal aspects of the teacher's representation that are crucial for the task.

- **Provides Effective Learning Signals in a Constrained Space**: The parameter space of an ONN is tied to physical device properties (e.g., phase shifts in MZIs, properties of optical materials). Optimizing these parameters solely with a downstream task loss can be highly challenging. KD provides rich, dense, intermediate supervisory signals (the teacher's feature maps) at various points in the network. This guidance makes the optimization problem more tractable, helping the student ONN navigate its constrained parameter space to find effective solutions that align with the proven representations of the electronic teacher.

### A.3 MORE EXPERIMENTS

This section provides a more comprehensive set of experimental results, including performance on the nuScenes dataset and further ablation studies.

Table 3: Detection Performance on nuScenes Validation dataset.

| Method | Setting | $AP_{50}$ | $AP_{75}$ | AP@[.5:.95] | Relative SOTA (%) | Recall@[.5:.95] |
|---|---|---|---|---|---|---|
| | Baseline | 41.2 | 17.8 | 20.3 | 100% | 46.4 |
| RegNet-Y | Hybrid-1 $\times layer$ ONN | 38.3 | 16.8 | 18.9 | 93.1% | 43.7 |
| | Hybrid-2 $\times layer$ ONN | 33.8 | 13.8 | 16.2 | 79.8% | 41.2 |
| | Hybrid-3 $\times layer$ ONN | 23.7 | 8.1 | 10.4 | 51.2% | 34.1 |
| | Distill-1 $\times layer$ ONN | **40.0** | **17.9** | **19.8** | **97.5%** | **46.2** |
| | Baseline | 52.5 | 24.5 | 27.1 | 100% | 52.7 |
| ViT-Base | Hybrid-1 $\times layer$ ONN | 51.1 | 22.2 | 25.3 | 93.4% | 50.8 |
| | Hybrid-2 $\times layer$ ONN | 46.7 | 19.0 | 22.5 | 83.0% | 49.2 |
| | Distill-1 $\times layer$ ONN | **52.0** | **23.9** | **26.3** | **97.1%** | **51.6** |

**Main results on nuScenes.** To further evaluate the generalizability of our framework, we conduct experiments on the nuScenes dataset, a large-scale benchmark for autonomous driving containing

diverse and complex outdoor scenes, shown in Table 3. Using the same hybrid backbone configurations as in COCO, we evaluate performance with 1–3 stages replaced by ONN modules. As shown in Table 3, the results show that our framework achieves comparable precision to the electronic baseline, with significant gains in inference speed and energy efficiency.

Table 4: Performance of Different Non-linear Functions in ONN module on COCO dataset.

| Method | Setting | $AP_{50}$ | $AP_{75}$ | AP@[.5:.95] | Recall@[.5:.95] |
|---|---|---|---|---|---|
| | Baseline | 47.1 | 32.5 | 30.7 | 67.9 |
| RegNet-Y | abs + learning bias | 40.8 | 26.9 | 25.8 | 63.8 |
| | \|abs\|$^2$ | 41.1 | 27.0 | 25.9 | **64.8** |
| | \|abs\|$^2$ + learning bias | **41.4** | **27.2** | **26.2** | 64.6 |

**Ablation on ONN Activation Functions.**  Due to the physical constraints of optical hardware, implementing nonlinear activation functions within ONNs is non-trivial and must conform to the underlying physics of light propagation. We conduct experiments using a variety of nonlinearities that are either physically realizable or approximable in optical systems, including absolute value ($|x|$, squared magnitude ($|x|^2$), and learned bias-shifted activations, the results as shown in Table 4. Among these, we observe that $|x|^2$ and **learned bias activations** consistently yield the best performance. Importantly, these nonlinearities functions also have strong physical relevance in optical systems:

- The **absolute function** ($|x|$) represents taking the *amplitude of the optical field* while discarding its phase. This simplification is common in optical systems where only intensity (not phase) is directly measurable.

- The **squared magnitude function** ($|x|^2$) corresponds to converting the *complex amplitude of a light field* into the measurable *light intensity* captured by detectors. This is the core operation in optical sensing and makes $|x|^2$ a naturally compatible nonlinearity for ONNs.

This alignment between mathematical formulation and optical measurability ensures that our design is not only effective in performance but also practical for real-world ONN hardware implementation. These results suggest that simple, physically grounded nonlinearities, when paired with our distillation framework, can provide sufficient expressiveness for object detection tasks without requiring complex or non-implementable optical operations.

Table 5: Performance of Number of Blocks per Stage.

| Method | Setting | $AP_{50}$ | $AP_{75}$ | AP@[.5:.95] | Recall@[.5:.95] |
|---|---|---|---|---|---|
| | Baseline | 47.1 | 32.5 | 30.7 | 67.9 |
| RegNet-Y | 1 | 39.6 | 25.9 | 24.8 | 63.5 |
| | 2 | **41.4** | **27.2** | **26.2** | **64.6** |
| | 4 | 41.3 | 27.0 | 26.1 | 64.5 |

**Effect of ONN Layer Depth.**  In this experiment, we evaluate how the depth of each ONN layer, measured by the number of stacked optical blocks, affects performance. Specifically, for each ONN stage replacing a portion of the backbone, we test using $[1, 2, 4]$ optical blocks in sequence, where each block consists of an optical modulation layer and a simple physically realizable nonlinearity (e.g., $|x|^2$ or bias-shifted activation), the results presented at Table 5.

- **1 Block**: This configuration minimizes optical depth, reducing latency but limits the capacity of the optical feature extractor.

- **2 Blocks**: Offers a balance between depth and learnability while remaining efficient and physically feasible.

- **4 Blocks**: Increases representation power but can amplify optical noise and lead to diminishing returns.

We observe that using **2 blocks per ONN layer achieves the best performance**, offering a strong trade-off between feature expressivity and optical robustness. The performance of 1 block setting is not satisfactory, while the 4-block setup shows slight performance drop due to accumulated phase noise and weaker generalization under hardware variability. These results suggest that **moderate ONN depth is optimal in hybrid systems**, where the goal is not only accuracy but also energy efficiency and robustness to physical noise.

Table 6: Ablation of different output channels on nuScenes.

| Method | Setting | $AP_{50}$ | $AP_{75}$ | AP@[.5:.95] | Recall@[.5:.95] |
|---|---|---|---|---|---|
| | Baseline | 41.2 | 17.8 | 20.3 | 46.4 |
| RegNet-Y | [288, 256] | 38.3 | 16.8 | 18.9 | 43.7 |
| | [288, 128] | 39.0 | 16.9 | 19.2 | 43.5 |
| | [288, 32] | **39.3** | **17.4** | **19.5** | **43.7** |
| | [288, 16] | 39.1 | 17.0 | 19.3 | 43.3 |

**Ablation experiments of output channels.**    We investigate how variation of the number of input and output channels in ONN modules affects detection performance. The impact of the output channels of ONN modules for Hybrid-$1 \times layer$ ONN is shown in Table 6. "Setting" column represents the input and output channels of the ONN module. Surprisingly, we find that fewer channel numbers of 32 output channels get the best performance.

Table 7: Detection Performance by adding noise to the model on MS COCO.

| Method | Setting | $AP_{50}$ | $AP_{75}$ | AP@[.5:.95] | Recall@[.5:.95] |
|---|---|---|---|---|---|
| RegNet-Y | Baseline | 46.5 | 31.2 | 29.8 | 67.0 |
| | Hybrid-$1 \times layer$ ONN | 39.6 | 26.0 | 24.9 | 63.3 |
| | Rodom noisy | 39.5 | 25.6 | 24.7 | 63.1 |

**Robustness of Noisy.**    To evaluate the robustness of our distillation-based framework, we inject uniform distribution noise into the ONN modules during the training phase, simulating real-world optical imperfections such as diffraction error, sensor noise, and analog drift, as shown in Table 7. Despite these disturbances, hybrid model maintains stable performance, demonstrating improved tolerance to noise and enhanced generalization. This confirms that distillation not only improves feature expressivity, but also acts as an implicit regularizer for noisy hardware.

Table 8: Detection Performance of different distillation strategies on nuScenes dataset.

| Method | Setting | $AP_{50}$ | $AP_{75}$ | AP@[.5:.95] | Relative SOTA (%) |
|---|---|---|---|---|---|
| RegNet-Y | Baseline | 41.2 | 17.8 | 20.3 | 100% |
| | Single-stage | 40.0 | 17.9 | 19.8 | 97.5% |
| | Two-stage | 38.7 | 16.5 | 18.8 | 92.6% |

**Different distillation strategies.**    We further explore the impact of different distillation strategies on the performance of the hybrid photoelectronic network, focusing on two paradigms of single-stage and two-stage strategies as follows, and the experiments presented in Table 8:

- **Single-stage:** For the student model training, we employ a distillation approach that leverages both backbone features and head logits from the teacher detector as supervisory signals. This ensures comprehensive knowledge transfer while maintaining efficiency.
- **Two-stage strategy:** The two-stage distillation pipeline first aligns the student's backbone features with those of the teacher detector, focusing solely on backbone-level knowledge transfer, ensuring effective feature learning while maintaining computational efficiency.

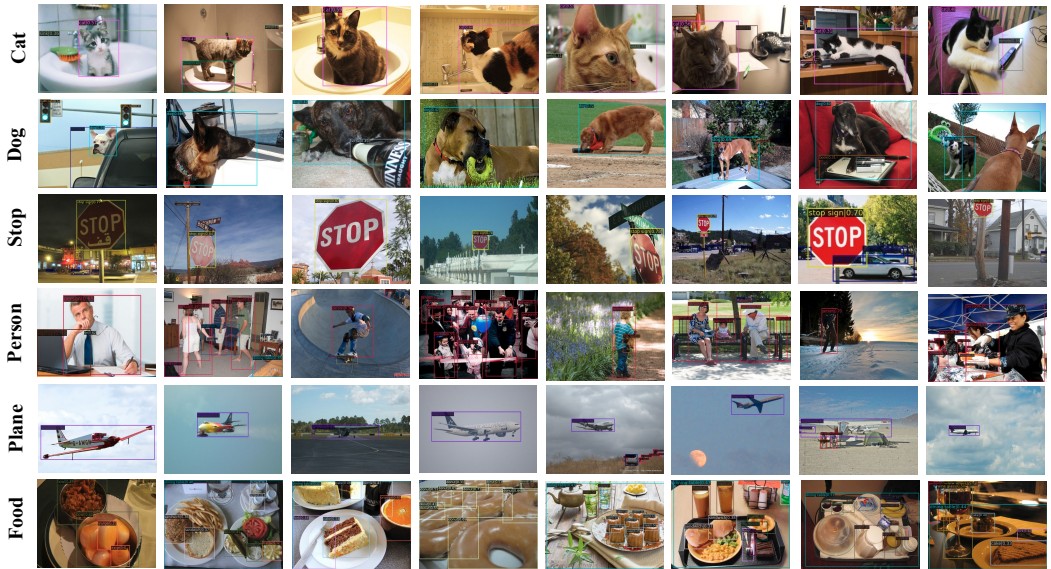

Figure 8: Detection visualization of our framework on the COCO dataset.

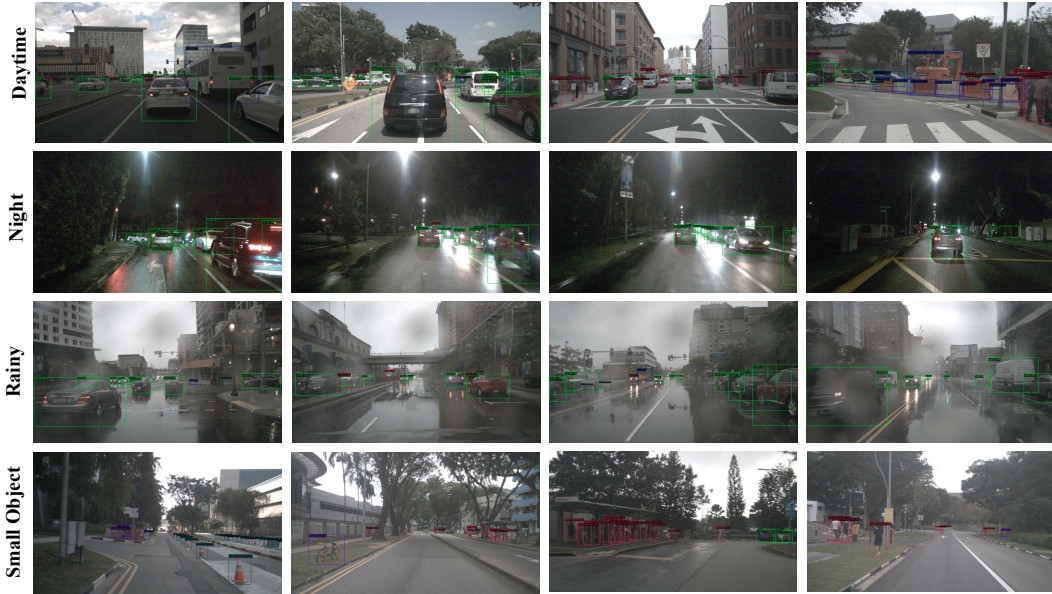

Figure 9: Detection performance of our framework on the nuScenes dataset.

## A.4 VISUALIZATION AND ANALYSIS

We provide additional qualitative visualizations for both COCO and nuScenes:

**COCO visualization** The Figure 8 presents qualitative detection results from our top-performing hybrid model (Hybrid-1 + Distill) on a diverse set of images from the MS COCO dataset. The examples show performance across various scenes, object densities, and scales, which demonstrate robust performance across challenging scenarios, including varying object scales, complex occlusions, and crowded scenes, highlighting the model's generalization capability. Notably, the results exhibit precise localization and high classification confidence even for small or partially obscured objects, validating the effectiveness of our approach.

**nuScenes visualization** Figure 9 presents qualitative detection results on challenging scenes from the nuScenes validation set. These examples highlight the model's performance in diverse conditions,

including varying weather, lighting (day/night), and complex urban environments, showcasing detections for key autonomous driving object classes.

