# OpenReview forum: "PHOENIX: Photonic Distillation Transfers Electronic Knowledge to Hybrid Optical Neural Networks"
_ICLR.cc/2026/Conference — Submitted to ICLR 2026_

### Official Review · Reviewer_osnR · 2025-11-01

**Soundness:** 3
**Presentation:** 3
**Contribution:** 3
**Rating:** 4
**Confidence:** 3

**Summary:**

The paper presents a novel hybrid optoelectronic neural network framework designed to bridge electronic deep learning and photonic hardware. The authors introduce PHOENIX, which integrates Optical Neural Networks (ONNs) into complex object detection pipelines through a knowledge distillation mechanism that transfers feature-level and logit-level knowledge from electronic “teacher” models to optical “student” networks. This method mitigates the limitations of ONNs - namely, limited nonlinearity and analog noise susceptibility by combining electronic early-stage feature extraction with photonic late-stage inference.

PHOENIX achieves impressive efficiency gains: up to 11.3x faster inference, 72.6% lower energy consumption, and 93% of baseline detection accuracy on large-scale datasets such as MS COCO and nuScenes. The framework generalizes across CNN-based (RegNet-Y) and Transformer-based (ViT-Base) backbones. Experimental validation and physical modeling (e.g., Rayleigh–Sommerfeld diffraction equations, FDTD simulations) substantiate the feasibility of the proposed hybrid architecture.

**Strengths:**

1. The idea of using electronic-optical knowledge distillation to compensate for ONN nonlinearity is conceptually innovative.


2. The work provides rigorous physical modeling (Rayleigh-Sommerfeld diffraction) and a detailed co-design framework unifying optical and electronic components.


3. Results across COCO and nuScenes show consistent performance retention and dramatic efficiency improvements.


4. The latency, FLOPs, and power breakdown analyses demonstrate careful hardware–software co-optimization.


5. PHOENIX is shown to be compatible with both CNN and Transformer architectures, broadening its applicability.

**Weaknesses:**

1. While the simulation and hardware parameters are grounded in literature, no empirical hardware prototype is demonstrated. A small-scale optical testbed result would significantly strengthen the claims.


2. The paper focuses on MSE and KL-based distillation losses; exploring attention transfer or relational distillation may further improve ONN performance.


3. The authors claim framework generality but do not analyze how ONN layer count or resolution scaling impacts photonic alignment errors or calibration overhead.


4. The noise robustness analysis is limited to phase noise. Evaluating under illumination or temperature drift conditions could add practical value.

**Questions:**

1. Could the authors provide empirical hardware measurements or even partial prototype validation to corroborate simulation-based latency and energy claims?


2. How does the optical-electronic interface bandwidth (modulator-detector pair) affect throughput in practice?


3. Have the authors explored alternative distillation objectives, such as feature contrastive or mutual-information-based methods, to further enhance ONN nonlinear expressivity?


4. For scalability, what is the expected error accumulation behavior when stacking multiple ONN layers (beyond three), given fabrication imperfections?


5. Can PHOENIX be extended to sequence or multimodal tasks (e.g., vision-language models), or is the framework limited to 2D visual inputs due to diffraction constraints?

---

### Official Review · Reviewer_hqQA · 2025-11-01

**Soundness:** 1
**Presentation:** 1
**Contribution:** 1
**Rating:** 2
**Confidence:** 4

**Summary:**

This paper proposes PHOENIX, a hybrid optoelectronic neural network framework that employs knowledge distillation to transfer representations from an electronic teacher model to a photonic student model for object detection tasks. The authors claim state-of-the-art performance on COCO and nuScenes datasets, with significant gains in energy efficiency and inference speed. However, the approach relies heavily on idealized optical simulations without physical validation, and the distillation method itself lacks novelty.

**Strengths:**

The work attempts to bridge the gap between optical neural networks and real-world computer vision tasks, which is a worthwhile direction.

**Weaknesses:**

- **Lack of novelty in distillation and optical modeling**: The knowledge distillation strategy is a straightforward application of feature mimicry between electronic and optical models, without introducing new distillation mechanisms or optical nonlinearities.

- **No physical validation**: The entire optical system is simulated under ideal conditions. There is no evidence that the proposed ONN can be realized in hardware or that it would perform similarly under real-world noise and constraints.

- **Overstated claims**: The paper makes strong assertions (e.g., “first deployment framework,” “universal,”) that are not fully supported by the methodology or results.

- **Theoretical implausibility**: The success of distilling highly nonlinear electronic networks into largely linear optical models challenges established understanding of representational capacity in deep learning. The authors do not provide theoretical or empirical justification for why this should be possible.

**Questions:**

See Weaknesses

---

### Official Review · Reviewer_RaVY · 2025-11-01

**Soundness:** 3
**Presentation:** 2
**Contribution:** 2
**Rating:** 4
**Confidence:** 4

**Summary:**

The paper introduces a hybrid optoelectronic distillation framework that combines the computational speed and energy efficiency of optical neural networks (ONNs) with the strong representational power of electronic deep models such as RegNet and Vision Transformers. PHOENIX replaces selected modules in these networks with diffractive ONN blocks and uses a knowledge distillation strategy to transfer feature- and object-level knowledge from a pretrained electronic teacher to the optical student, enabling the optical modules to mimic electronic behavior despite their limited nonlinearity. Applied to large-scale object detection tasks like MS COCO and nuScenes, PHOENIX achieves up to 93% of the original model accuracy while delivering over 70% energy reduction and more than tenfold speedup. The work positions itself as the first general framework for large-scale, hardware-aware deployment of hybrid optical-electronic neural networks, offering a promising direction for fast, low-power, and physically interpretable AI systems.

**Strengths:**

The paper is original in proposing a unified framework, PHOENIX, that meaningfully integrates optical neural networks into mainstream computer vision pipelines through a knowledge distillation mechanism. While optical computing itself is not new, PHOENIX’s formulation of optoelectronic distillation, transferring learned representations from electronic teacher models to optical student modules, represents a creative and practical bridge between two previously disconnected paradigms. The work also extends ONN research beyond small-scale classification to industrial-scale object detection tasks like COCO and nuScenes, which is a notable and novel application domain for photonic computation. In terms of quality, the methodology is carefully constructed, with a clear description of the hybrid architecture, theoretical modeling of light propagation, and a comprehensive experimental evaluation that demonstrates consistent performance recovery through distillation, alongside energy and latency analyses. The experiments are thoughtfully designed to validate both functional and hardware-level benefits.

**Weaknesses:**

1. Unspecified optical encoding and channelization (core systems gap). The paper never specifies how a high-dimensional feature tensor H×W×Cis encoded onto the optical carrier(s): whether channels are mapped via spatial tiling/segmentation, wavelength/polarization multiplexing, or time-division, nor how tensors are loaded into and read out from modulators/detectors. The only hint is a conceptual statement that convolution with arbitrary N_"in"  " ⁣"×" ⁣" N_"out" is achieved by “optical network units” and “segmentation of the effective optical field,” which leaves critical bandwidth/throughput and crosstalk questions unanswered. Add a full I/O & encoding spec: (i) chosen multiplexing mode(s) (spatial/wavelength/polarization/time), (ii) modulator pixel count & symbol rate, (iii) photodetector / ADC bandwidth & quantization, (iv) per-channel isolation/crosstalk tolerances, and (v) the mapping from H×W×Ctensors to optical degrees of freedom. Then report end-to-end throughput (fps) and SNR under that scheme.

2. Downsampling/stride left implicit when stages move to optics. RegNet stages perform stride-2 reductions; the paper replaces later stages with DONN modules but never shows how optical downsampling (e.g., stride-2 conv, pooling, or sub-sampling) is realized physically—or whether downsampling remains electronic at the ONN boundary. Table 2 lists strides and ONN replacement flags but provides no optical mechanism (masking, microlens decimation, Fourier-domain cropping/decimation, or detector-plane sampling geometry) for stride-2. Specify one of: (a) an optical stride-2 design (e.g., coded aperture + detector binning, or Fourier-patch decimation) with tolerance analysis; (b) an explicit decision to keep all downsampling electronic with diagrams of the E/O boundary tensors; (c) a hybrid where downsampling is implemented by detector-array sampling pitch. Then validate with ablations showing mAP vs. optical vs. electronic stride.

3. Latency model assumes perfect parallelism across channels and spatial positions. The stage latency is computed as t_"EOM" +t_"ONN" +t_"PD" (≈10.4 ns) for a single pass, with a 10^5× “component-level” speedup and 11.3× system speedup, but does not account for (i) channel serialization if channels are time-multiplexed, (ii) multiple E/O load cycles per feature map, (iii) detector integration time vs. SNR, or (iv) pipeline bubbles between electronic and optical segments. Provide a throughput-aware latency model that multiplies the single-pass time by the number of required channel/time/wavelength passes (or, if fully parallel, justify the hardware parallelism and area). Report frames/sec for a concrete tensor (e.g., C=256, H=W=56) under the stated encoding.

**Questions:**

1.	 How is the high-dimensional feature tensor (H × W × C) encoded into the optical domain? Please clarify whether channels are mapped spatially, by wavelength, or time-multiplexed, and how data transfer and synchronization are handled across E/O and O/E interfaces.

2.	How is downsampling (e.g., stride-2 convolution) implemented in the optical domain once RegNet or ViT stages are replaced by DONN modules? Is it performed optically or still handled electronically?

3.	Does the reported energy analysis include the power consumption from data movement, and control electronics? If not, how would including those affect the claimed 72–86% energy savings?

4.	How does the ONN module handle multiple feature channels—are they processed fully in parallel or reused sequentially? If reused, what is the resulting latency per feature map and the real achievable speedup?

---

### Official Review · Reviewer_9osT · 2025-11-02

**Soundness:** 3
**Presentation:** 3
**Contribution:** 2
**Rating:** 2
**Confidence:** 4

**Summary:**

This paper proposes a optoelectrically fused neural network deployment framework PHOENIX that replaces later CNN/ViT stages with ONN modules and uses teacher–student distillation to recover accuracy.

**Strengths:**

Strengths:
* Clear, readable system diagram and training recipe;the pipeline in Fig. 2 is easy to follow
* Work on a larger problem in industrial-level large datasets (e.g., COCO) and benchmark models. Demonstarte the use of optics in relastic AI problem.

**Weaknesses:**

Weaknesses (core novelty concerns):
* My main concern is the poor tech novelty of this work:  The “hybrid” is a stage-swap: early electronic stages followed by diffractive ONN stages, which is not new and has been used on optical-electrical hybrid system to imporve performance
* Moreover, no methodological novelty in distillation. The KD recipe is conventiona, feature L2 + logit KL, while this is a key contribution claimed by the author.
* Still a huge accurcay gap and concern on real challenging task? The baseline accurcay has been 100%, which means the task is not hard anymore, while the accuracy drop as ONN depth grows, with a clear gain to the digital baseline. E.gt., 88.3% vs 100% and 93.0% vs 100%.

**Questions:**

Besides weakness, I have more questions:
* What is new in the distillation beyond Eq. (3)?

---

### Meta-Review · Area_Chair_CTim · 2026-01-08

**Summary:**

The consensus across reviews is that the paper at its current form is with substantial concerns about novelty, technical completeness, and evidentiary strength. While reviewers acknowledge the ambition of moving ONNs beyond toy benchmarks and recognize the clarity of the high-level framework and experimental setup, multiple critical issues emerge: the core architectural idea is widely viewed as an incremental stage-swap hybridization rather than a fundamentally new methodology; the distillation strategy is standard and not convincingly justified as a novel contribution; and key system-level details about optical encoding, channelization, downsampling, latency, and energy accounting are missing or idealized, undermining the credibility of the claimed efficiency gains. The lack of any hardware validation, combined with strong claims about generality and deployment readiness, exacerbates concerns about overstatement and theoretical plausibility, particularly given the limited nonlinearity of ONNs.

No rebuttal was provided, leaving major reviewer questions unanswered and preventing clarification or mitigation of these weaknesses. The appropriate decision is rejection, primarily due to insufficient novelty, unaddressed technical gaps, and the absence of rebuttal to resolve substantive concerns.

**Reviewer Concerns:**

No rebuttal is provided, leaving major reviewer questions unanswered.

**Reviewer Scores:**

Two reviewers recommend rejection with low soundness and contribution scores. The remaining reviewers rate the paper as below the acceptance threshold despite some positive aspects.

---

### Decision · Program_Chairs · 2026-01-26

Reject